# Valoration of the Synthetic Antioxidant Tris-(Diterbutyl-Phenol)-Phosphite (Irgafos P-168) from Industrial Wastewater and Application in Polypropylene Matrices to Minimize Its Thermal Degradation

**DOI:** 10.3390/molecules28073163

**Published:** 2023-04-02

**Authors:** Joaquín Hernández-Fernández, Heidis Cano, Ana Fonseca Reyes

**Affiliations:** 1Chemistry Program, Department of Natural and Exact Sciences, San Pablo Campus, University of Cartagena, Cartagena 130015, Colombia; 2Chemical Engineering Program, School of Engineering, Universidad Tecnológica de Bolivar, Parque Industrial y Tecnológico Carlos Vélez Pombo, Km 1 Vía Turbaco, Turbaco 130001, Colombia; 3Department of Natural and Exact Science, Universidad de la Costa, Barranquilla 30300, Colombia; 4Department of Civil and Environmental, Universidad de la Costa, Barranquilla 080002, Colombia; 5Department of Mechanical Engineering, Universidad del Norte, Barranquilla 081007, Colombia

**Keywords:** valoration, synthetic antioxidant, Irgafos P-168, industrial wastewater, polypropylene, thermal degradation

## Abstract

Industrial wastewater from petrochemical processes is an essential source of the synthetic phenolic phosphite antioxidant (Irgafos P-168), which negatively affects the environment. For the determination and analysis of Irgafos P-168, DSC, HPLC-MS, and FTIR methodologies were used. Solid phase extraction (SPE) proved to be the best technique for extracting Irgafos from wastewater. HPLC-MS and SPE determined the repeatability, reproducibility, and linearity of the method and the SPE of the standards and samples. The relative standard deviations, errors, and correlation coefficients for the repeatability and reproducibility of the calibration curves were less than 4.4% and 4.2% and greater than 0.99955, respectively. The analysis of variance (ANOVA), using the Fisher method with confidence in 95% of the data, did not reveal significant differences between the mentioned parameters. The removal of the antioxidant from the wastewater by SPE showed recovery percentages higher than 91.03%, and the chemical characterization of this antioxidant by FTIR spectroscopy, DSC, TGA, and MS showed it to be structurally the same as the Irgafos P-168 molecule. The recovered Irgafos was added to the polypropylene matrix, significantly improving its oxidation times. An OIT analysis, performed using DSC, showed that the recovered Irgafos-blended polypropylene (PP) demonstrated oxidative degradation at 8 min. With the addition of the Irgafos, the oxidation time was 13 min. This increases the polypropylene’s useful life and minimizes the environmental impact of the wastewater.

## 1. Introduction

Industrial processes are some of the productive forces that have significantly contributed to the development of today’s society, developing products and providing services that have improved our quality of life [1,2,3]. However, one of the most significant problems revolving around industrial production is the pollution that it generates, since in every industrial process and each of its unit operations, specific residues are developed, most of which are toxic and highly polluting [2,4,5]. One of the products whose highest consumption has increased in the last century is plastic [6], thanks to its durability and great versatility [7]. With this increase in production come several industry challenges related to waste treatment [7]. The petrochemical industry is a producer of plastics or polymers, and the industry uses additives to increase or improve the properties of the plastic, such as increasing its useful life, lubricating the plastic, and increasing its resistance. Additionally, some plastics are non-slip, antistatic, or antioxidant [3,8,9]. All these additives, which are added in the manufacturing process, are not fully absorbed by the polymer and become process residues. Due to the dynamics of these processes, these residues are usually discharged into the wastewater, which may result in their discharge into different bodies of water [4,10,11]. These wastewaters usually have an unusually high chemical oxygen demand (DCO) and biological oxygen demand (BOD) because they contain some volatile organic compounds, phenols, minerals, polycyclic aromatics, and many others. Conventional treatment processes are not efficient or effective in these wastewaters [12,13], resulting in the presence of these chemical compounds in the aquatic environment. Although most of these additives can indeed be present in low concentrations, it is also correct that these low concentrations are already toxic enough to cause problems for both health and biodiversity [14,15]. The plastic industries that can dump this waste are characterized by their use of multiple additives that prevent the oxidation of their resins, which are synthesized by exposure to light or heat [16]. This oxidation negatively affects the polymers since their physical–mechanical capacities are significantly reduced. To avoid this, a family of additives called antioxidants (AO) [17,18] is used. PP is one of these polymers. PP has gained great importance in the industry since, thanks to its low density and high strength, it can be heated and cooled without losing any of its characteristics, such as its elevated mechanical and chemical stability [19,20,21,22,23,24,25,26,27]. Polypropylene is also a product of great economic interest since it is versatile in its application in packaging, textiles, household appliances, toys, and medical products, etc. [20,28,29,30,31,32]. For this reason, this product is in great demand (79 million tons in 2011) [33]. It may bring associated with certain environmental risks [34,35], since OAs are used in its production to avoid the oxidative degeneration of the polymer. OAs, such as Irgafos P-168, are used to improve the thermal stability and prolong the useful life of these polymers [36,37]. Irgafos 168 is manufactured by several companies worldwide, such as BASF, Toronto Research Chemicals, Spex, and Cymit Chemicals, among many others. Irgafos P-168 is a compound that acts as an oxidation inhibitor and helps to protect the polymer from the damaging effects of heat and light [37]. This protection, which is conferred to the polymers by OAs, makes them necessary in the polypropylene production processes, especially in the extrusion stages in which the polymer resins are dosed and mixed. However, some of these additives are not absorbed during the process: they are removed during the polypropylene washing and cleaning stages and are later discarded in wastewater and eventually in bodies of water [38]. Irgafos 168, which may be present in this discharge, is a compound with a chemical structure comprising an organophosphate (see Figure 1) attached to three rings of phenols that can be toxic to aquatic organisms and can affect water quality [39,40,41,42]. Irgafos P-168 can break down into potentially harmful secondary compounds [26,43,44,45]. Research on the health impact of Irgafos P-168 is not fully defined; however, the degradation products of Irgafos P-168, such as di-tertbutyl-phenol and tris(2,4-di-tert-butylphenyl) phosphate, are toxic. Some regulations have established their maximum concentrations between 5 and 40 mg/L. One study found concentrations of these Irgafos P-168 derivatives at 400 mg/L or more in bottles made with polymers that contained this additive [46,47,48,49,50]. In bodies of water, Irgafos P-168 can become hydrolyzed over time, and under various environmental conditions it can form di-terbutyl-phenol, which is highly toxic [51].

This di-terbutyl-phenol derived from Irgafos P-168 is cytotoxic, and it directly and significantly increases the expression of the P53 gene in more than one cell line. In rats, it caused hepatic and renal toxicity, increased organ weight, and histopathological changes [52]. For these rodents, the no-harm concentrations were between 5.01 and 20.03 mg kg^−1^ day^−1^. Irgafos P-168 (tris(2,4-di-tert-butylphenyl) phosphite) has direct effects on human cells, such as preventing their growth, and can cause endocrine disruption [51,53]. The impact of these molecules on health and the environment has been determined thanks to the existence of analytical quantification techniques and previous sample treatments. Irgafos P-168 has been effectively extracted using extraction techniques such as solid phase microextraction (SPME), liquid–liquid extraction (LLE), supercritical fluid extraction (SFE), ultrasonic-assisted extraction (UAE) [54], solvent microextraction (MEPS), pressurized liquid fluid extraction (PLE) and dispersive solid phase extraction (dSPE) [55]. Some of the techniques used to quantify Irgafos P-168 in different processes and under other conditions have included mass spectrometry [56], pyrolysis-GC/MS [51], FTIR spectroscopy [57,58], DSC and TGA [59], NMR and UV–Vis [58], and HPLC [60,61,62].

In this investigation, we will evaluate the reliability of the Irgafos P-168 measurement method using HPLC. For this purpose, the repeatability, reproducibility, linearity, and recovery percentage are evaluated. The Irgafos molecule is quantified in industrial wastewater samples from polypropylene production plants. In this research, a methodology for extracting Irgafos from wastewater is proposed to minimize its environmental impact. The extracted Irgafos is purified and characterized using mass spectrometry, FTIR spectroscopy, TGA, and DSC. To guarantee the performance of this recovered Irgafos, it is added to a virgin PP matrix, and the resulting mixture is evaluated using FTIR spectroscopy, TGA, OIT and MFI.

## 2. Materials and Methods

### 2.1. Recovery of the Additive

#### 2.1.1. Samples Collection

Samples were acquired at a polypropylene production facility that divides the production of the material into four steps (Figure 2): arrival, polymerization, additivation, and granulation, which are described in eight steps. In step 1, the material is received; in step 2, the raw material enters the reactor. It is mixed with a catalyst (Ziegler–Natta) that reduces the activation energy for the polymerization reaction. In step 3, darkening the extrusion, in order to lower the temperature and produce a more uniform grain, water is employed in the polymer extrusion process. After the polymer is produced, it is transferred to a purging column (step 4) for purification. The additives that improve and complement the properties of the PP are then added (step 5). In step 6, the extrusion and pelletization of the PP mixture containing the additives occur. The samples of interest in this inquiry primarily originated from the condensates of the extrusion and deodorization procedures (step 6). Every 12 min for 4 h, 200 mL of samples were taken, and the samples were kept at 4 °C in an amber glass bottle.

#### 2.1.2. Extraction System for Sample and Standard

The collected sample was subjected to filtration through a PTFE filter before being extracted. Solid phase extraction (SPE) was performed with Strata-X tubes (33 μm). A total of 15 mL of the sample was used, and the operating flow rate was 1.1 mL min^−1^. The SPE conditioning was carried out with methanol and distilled water with an 80:20 ratio of MeOH: H_2_O. To carry out the elution of OA in SPE, 10 mL of acetonitrile (ACN) was used as a solvent. The eluted extract from the SPE was dissolved in ACN to 1 mL and separated by HPLC. When 500 mL of the extract was obtained, the pre-concentration process was repeated. The recovered solid was then dried (by recovering it with a stream of N_2_) and subjected to FTIR spectroscopy, DSC, TGA, and HPLC-MS analysis. The obtained export product (Irgafos P-168) was stored in a controlled environment for later use in virgin PP resin. Figure 3 represents the general outline of the SPE process. In the SPE process, the residual water sample, which contained several residues of additives, such as Irgafos P-168 and even trace metals, was poured into the extraction cartridge. The extraction cartridge included a porous solid phase which, given its chemical structure, has a specific molecular region for non-polar substances. Although Irgafos has polarized regions in its chemical structure, the three Di-tert-butyl groups it has generate a steric hindrance, so the polar region does not have an affinity. This causes steric hindrance, so it has an affinity with the stationary phase of the extraction capsule.

#### 2.1.3. Separation, Identification, and Characterization System by HPLC/DAD/MS/MS/MS

This analysis was conducted using a Micromass Quattro II triple quadrupole mass spectrometer and an Agilent 1200 HPCL. The system includes a Lichrosorb RP-18 column (4.6 m × 200 mm × 5 µ), syringes of 5 and 10 mL, a precision balance, a degasser (G1322A), a pump (G1311A), an automatic sampler (G1313A), a column carrier (G1316A), and these components. The chromatographic modifications were made using the Irgafos P-168 solution, which was produced in ACN, as a starting point. The solvents ACN and H_2_O were combined in the following ratios to form the mobile phase: 83 and 17% (1 min, 15 mL/min); 93 and 7% (2 min, 2 mL/min); 95 and 5% (3.5 min, 3.5 mL/min); and 100 and 0% (8 min, 3.5 mL/min). The temperature in the column was 50 °C. Irgafos 168 was identified using MS and MS/MS fragments.

### 2.2. Reincorporation of Wastewater Additive in the PP Production Process

A virgin PP resin with no additives was used. It was separately combined with the recovered PP-Irgafos 168 samples and pure PP-Irgafos 168 samples to produce recovered PP-Irgafos 168 samples. The PP powder was premixed with 0.1 weight percent of recovered Irgafos P-168 and neat Irgafos 168. An ordinary Prodex Henschel 115JSS mixer was used for this process, running at 800 revolutions per minute for 7 min. Melt extrusion was used for mixing with a Welex-200 24.1 extruder. (see Figure 4) The extruder has five heating zones that guaranteed the correct homogenization of the PP and the additive mixtures. The working temperatures of the extruder were 190, 195, 200, 210, and 220 °C.

The PP solid produced by the extruder die was granulated, and these samples were spiked with 1000 mg kg^−1^ of recovered Irgafos P-168 and 1000 mg kg^−1^ of pure Irgafos P-168. Irgafos P-168 was diluted in ACN to a 500 mg/L to create a standard, and 5 mL of this solution was then dissolved in 100 mL to create a 25 mg/L solution.

### 2.3. Sample Evaluation

#### 2.3.1. Heat Flux Characterization System (DSC)

To ascertain the samples’ oxidation induction time (OIT), a DSC Q2000 V24.11 Build 124 device was used for calorimetric analysis. Results were obtained using a 6.1 mg sample under nitrogen and oxygen ambient conditions. The experiment was conducted to investigate the material’s oxidation and volatility effects. The sample was first heated isothermally for 5 min at 60 °C and then for 20 min at 200 °C in a nitrogen atmosphere moving at 50 mL/min. The atmosphere was then altered to 50 mL/min of airflow and 30 min of oxidation at 200 °C. It was feasible to determine the OIT value, which corresponds to the instant at which the change in pitch happens, due to this change in atmosphere’s revelation of a difference in the slope of the exothermic heat.

#### 2.3.2. Spectrometric Characterization System

For the spectrometric characterization, Nicolet 6700 reference infrared spectroscopy equipment was used, with readings between 400 and 600 cm^−1^ and a resolution of 2 cm^−1^ (reflection mode).

#### 2.3.3. Thermogravimetric Characterization System

Using a Perkin Elmer TGA7 thermobalance and a N_2_ environment flowing at a rate of 50 mL/min, a TGA was conducted. The temperature at which there was a 5% mass loss served as the benchmark for the initial degradation temperature, and the DTG curve served as the benchmark for the maximum degradation temperature.

#### 2.3.4. Melt Flow Index (MFI)

A Tinius Olsen MP1200 plastometer was used to determine the MFI. The working temperature inside the equipment cylinder was 230 °C, and a 2.16 kg piston was used to move the molten material.

## 3. Analysis and Discussion

### 3.1. Standards Calibration Curve for Irgafos P-168

We proceeded to establish the repeatability and reproducibility of the quantification method in which the concentration of the Irgafos P-168 recovered using the HPLC-MS method would be measured. For this, a calibration curve was generated with known amounts of concentrations between 0 and 5000 ppm. To carry out this calibration and guarantee its reliability, seven different samples were analyzed for five days by the same analyst. The tests were also carried out with five various analysts, each analyzing seven samples in different concentrations, as shown in Table 1. The maximum relative standard deviations for repeatability and reproducibility were 3% and 4.4%, respectively, and their respective maximum errors were 4.2% and 2.6%.

The acceptance value for validating the calibration of the deviation obtained with respect to the expected values was less than 15% [62]; in addition, an ANOVA was applied using the Fisher grouping method, with which it was possible to establish that the difference in the measurements was not significant. The reliability of the process was established with a confidence of 95%, and a significance value of α = 0.05 was established. Figure 2 shows the confidence intervals, comparing pairs of data columns for the same analyst (Figure 5a) and for different analysts (Figure 5b). These are the repeatability and reproducibility. The semi-continuous line perpendicular to all intervals indicates the position of zero in a break. When this semi-continuous line intersects with any of the intervals, it indicates that zero belongs to the gap, showing that there are no appreciable differences between the data pairs being examined. Therefore, it is concluded that there are no significant differences. Consequently, the method is repeatable and reproducible.

### 3.2. SPE of Standards with Acetonitrile

The values for the variables studied in the extraction are presented in Table 2. Here (Table 2), the maximum errors for repeatability and reproducibility were 4.8 % and 5.3 %, respectively, the maximum relative standard deviations of repeatability and reproducibility were 3.2 % and 2.5%, and the complete extractions were 96% and 98%.

Figure 6 shows the confidence intervals, comparing pairs of data columns for the same analyst (Figure 6a) and for different analysts (Figure 6b), i.e., the repeatability and reproducibility of the SPE. The significance of the difference for the intraday and interday data of the SPE was also analyzed using the Fisher method, with an *p* = 0.05 value and 95% confidence. Under the same analysis made in Section 3.1, Figure 6a,b, which show the repeatability and reproducibility of the standards extraction, respectively, show that there are no appreciable variances in the data.

### 3.3. Linearity Analysis for the Calibration Curve and SPE of Standards

Using Origin, the average interday and intraday data were plotted against theoretical concentrations for the calibration curve and SPE. Each underwent a linear regression to obtain the parameters of R^2^ and the correlation coefficient (r). Figure 7a,b show the linear regressions for the calibration curve, in which intraday and interday values of 0.99966 and 0.99997 were obtained for R^2^ and 0.99983 and 0.99999 were obtained for the correlation coefficient. Figure 7c,d provide the same information as the previous paragraphs for the SPE unemployment. Here, intraday and interday values of 0.99999 and 0.99985 for R2 and 0.99999 and 0.99993 for the correlation coefficient were obtained.

### 3.4. Analysis of Concentration and Recovery of Irgafos 168 in Samples

To analyze the recovered Irgafos P-168, 40 samples were taken over 40 consecutive days. These samples were subjected to analysis by five different analysts on the same day, and the data are listed in Table 3. From the data, we can see that the RSDs were less than 15%; more specifically, the maximum was 4.7%, and the concentration of the recovered samples was in all cases was greater than 91.03%. The highest concentration, which was taken on day 26, was 99.38%, with a value average of 2548 ppm.

In order to assess the model’s error, a linear regression was conducted. The results showed a proportionate association between the concentration determined by the analysts and the concentration of the Irgafos P-168 retrieved. As shown graphically in Figure 8, the model is predictive since it has an R^2^ of 0.99955 and a correlation coefficient of 0.99978.

### 3.5. FTIR Analysis of the Recovered Irgafos

Samples of Irgafos recovered from the wastewater were taken for analysis by FTIR spectroscopy. The results indicated that the recovered Irgafos P-168 presented significant similarities with the spectrum of the compound in its pure state, as shown in Figure 9, suggesting a high purity of the recovered Irgafos P-168. Some differences are noted in the spectra due to noise from the equipment and the low concentrations of the samples used in this equipment.

In analyzing the spectrogram graph corresponding to the pure and recovered Irgafos P-168, an absorption band is found between 3000 and 2800 cm^−1^ that is related to the alcohol and ester groups. Precisely at 2868 cm^−1^, a band of overlapping absorption corresponds to CH. At 2961 cm^−1^, there is another band corresponding to CH_3_. In the Irgafos P-168, the aromatic C=C bond is tenuously situated at 1602 cm^−1^ and 1491 cm^−1^. At 1398 cm^−1^, there is a corresponding deformed CH and CH_3_ absorption band. Finally, at 1212 cm^−1^, there is a band of the phosphite group (C-O-P. All these particularities are typical of Irgafos P-168 [63].

### 3.6. TGA Analysis of the Recovered Irgafos

The physicochemical properties of the recovered material were analyzed by measuring its thermal stability. To measure its thermal decomposition point, scanning thermography tests were carried out on the Irgafos P-168 that was recovered after use in the PP process. Temperature changes were applied to the recovered and pure Irgafos samples at a rate of roughly 0.3 °C/s. It was observed that the thermal resistance of both samples was quite similar since in both cases, the mass did not vary substantially until it reached 280 °C. After this point, it began to decay; the results obtained suggest that the recovery process of the Irgafos P-168 did not significantly affect its thermal stability as its thermal decomposition point remained similar to that of pure Irgafos P-168, as can be seen in Figure 10.

To achieve a clearer vision of the similarity of the data, the derivative of the previous data was graphed (Figure 11) to show the similarity of the data. In these graphs, it can be seen that both the recovered Irgafos P-168 and the pure Irgafos P-168 demonstrate a change in the inflection point of the thermal curve at the same value, approximately 425 °C.

Table 4 shows the mass of the ions and molecules obtained from the fragmentation of Irgafos P-168 by the detection technique of mass spectrometry coupled with TGA. Figure 12 shows the chemical and molecular structure of some of the ions generated from the fragmentation of the Irgafos P-168A molecular ion with a mass of 206; M+ can be seen in the 2,4-DTBP mass spectrum. The molecular ion at *m*/*z* 191 changes into an intense fragment ion after losing the methyl group. The phosphite mass spectrum shows the presence of the M+ molecular ion, and the loss of the 2,4-ditest group from the molecular ion is assumed to be butyl phenoxy, which produces the peak with a maximum intensity at *m*/*z* 441. It was then discovered that the sharp peak at *m*/*z* 647 in the phosphite spectrum is caused by a fragment ion created when the M+ molecular ion lost a methyl group. They contain the compound 2,4-di-tert-butylphenol in the phosphite spectrum at *m*/*z* 206. The fragment ions associated with the tropylium ion at *m*/*z* 91 and the tert-butyl fragment at *m*/*z* 57 are present in all ranges.

### 3.7. DSC Analysis of the Recovered Irgafos

The purity of a chemical compound such as Irgafos P-168 can be analyzed using various scientific techniques. In this case, DSC was used to assess the purity of the Irgafos recovered from wastewater. In the test carried out, a temperature change was applied at a rate of 19.7 °C/min. The data obtained indicated that the melting point of the sample was close to 187 °C, and it reached a maximum point of heat absorption of approximately 40 mW. When comparing these results with those obtained by subjecting the compound in its pure state to the same conditions, it was possible to conclude that the purity of the recovered Irgafos is still relatively high, given the similarity of the data (see Figure 13).

### 3.8. FTIR Analysis of PP Plus Pure Irgafos P-168 and PP Plus Recovered Irgafos P-168

To generate the most accurate analysis possible, the reuse of the recovered Irgafos P-168 in the production of polypropylene was carried out to compare the characteristics of the final material with both the pure and the recovered additive.

Both materials were subjected to FTIR spectroscopy studies (see Figura 14) with resultant graphs. We can see that the absorbance regions are the same for both materials, meaning that the performance of P-168 is high since the characteristics of the obtained PP are the same as those obtained with the pure additive. The observed spectra (Figure 14) display two bands between 1305 and 1055 cm^−1^ that correspond to the symmetric and asymmetric stretches of the ester group of Irgafos P-168 and a peak at roughly 1737 cm^−1^ that is suggestive of the ester group (C=O) present in the structure of the Irgafos P-168. Between 2953 and 2975 cm^−1^, one can see the typical CH_3_ group band. Irgafos P-168’s chemical composition also reveals these classes. Another indication of the existence of aromatic groups in the spectrum is the relative strength of the absorption, which is typical of the spectra of aromatic compounds and occurs in the range of 1450–1500 cm^−1^.

### 3.9. Melt Flow Index (MFI) and Molecular Weight Distribution

The thermal properties of polypropylene are some of the most critical qualities for its processing and the evaluation of multiple changes that can affect its quality during application. Using the MFI (see Figure 15), we can estimate the melt processing of the PP; the higher the MFI values, the higher the melt flowability, and the lower the MFI values, the lower the melt flowability of the PP. The MFI measurements of each sample were performed in triplicate. The MFI results are shown in Figure 15, and it can be seen that the MFI of the PP decreased with the dosage of the recovered P-168 and pure P-168. Concentrations of 0.1% of pure or recovered P-168 significantly improve the MFI. Without additives, PP has an MFI of 6.5. This MFI decreased to 4.1 and 4.3 with the addition of pure P-168 and recovered P-168, respectively.

This molecular weight was determined experimentally and theoretically using the Bramner equation as a reference (Equation (1)):(1)$$\bar{M_{v}}=-8480.6\times Ln\;\mathrm{MFI}+62,836$$

The Bramner equation shows the relationship between the MFI and the average molecular weight (Mw). In Figure 15, an inverse relationship between the MFI and Mw is observed. Samples with an MFI of 6.5, 4.1, and 4.3 g/10 min presented Mw values of 46,961, 50, 870, and 50,466 kDaltons. It was observed that the addition of Irgafos P-168 stabilized the PP, preventing its complete degradation, decreasing its fluidity, and increasing its molecular weight distribution.

### 3.10. OIT

To determine the oxidation time, the changes in the slope of the curve generated by the DSC concerning the expected time and the heat flux were taken into account. As shown in Figure 16, an endothermic peak was observed, but with time and the oxygen atmosphere, the slope of the curves changed, revealing a new exothermic behavior consistent with oxidation. It can be seen that the non-stabilized PP had a significantly shorter oxidation time than the PP with additives. The OIT value for the unstabilized PP was 0.9 min, taking the change in slope at 17.3 min as reference, and the OIT for pure PP + P-168 and recovered PP + P-168 were 8 and 7.8 min, respectively. This demonstrates how the presence of P-168 minimizes the oxidation processes of PP and improves its thermal stability due to its coupling in the polymeric matrix.

## 4. Conclusions

According to the findings of this study, it is possible to recover Irgafos P-168 with a high degree of purity in amounts greater than 91.03%, and its integration into the PP matrix enhances the material’s thermal and thermo-oxidative stability. Without thermal stabilizers, PP cannot be used since it would entirely deteriorate during the extrusion process. As a result, the recovery of this Irgafos P-168 at such high purity levels exemplifies a critical methodology for use in the industrial sector, promotes the use of sustainable raw materials, and makes a significant contribution to the circular economy of the PP polymer industry.

## Figures and Tables

**Figure 1 molecules-28-03163-f001:**
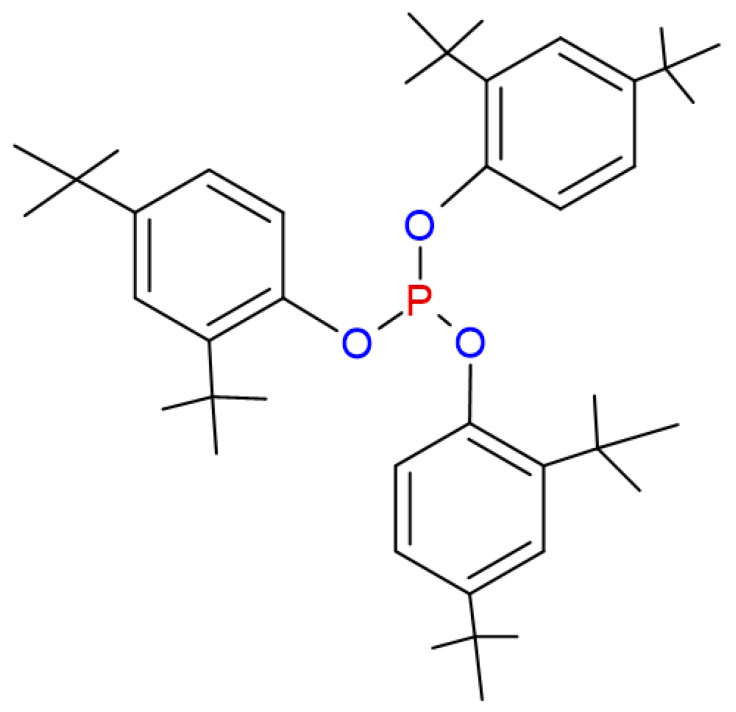
Chemical structure of Irgafos P-168.

**Figure 2 molecules-28-03163-f002:**
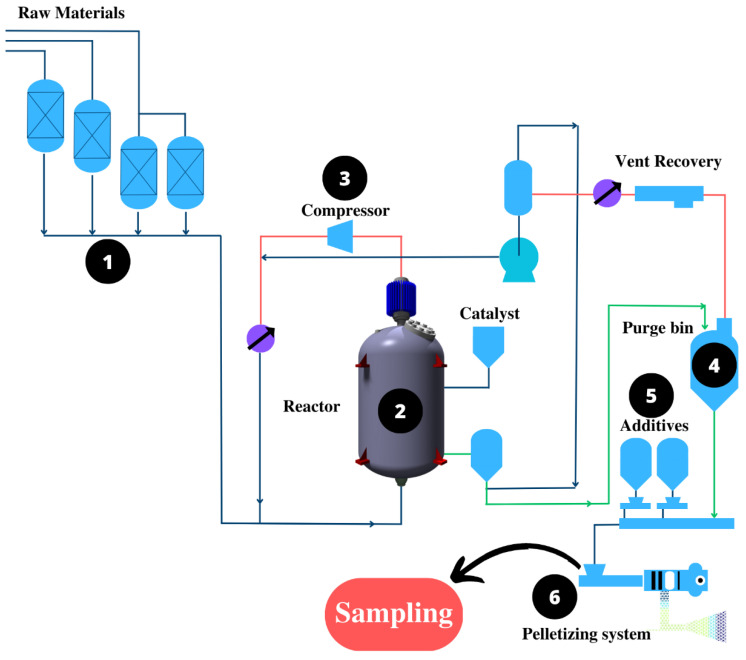
PP production flow diagram.

**Figure 3 molecules-28-03163-f003:**
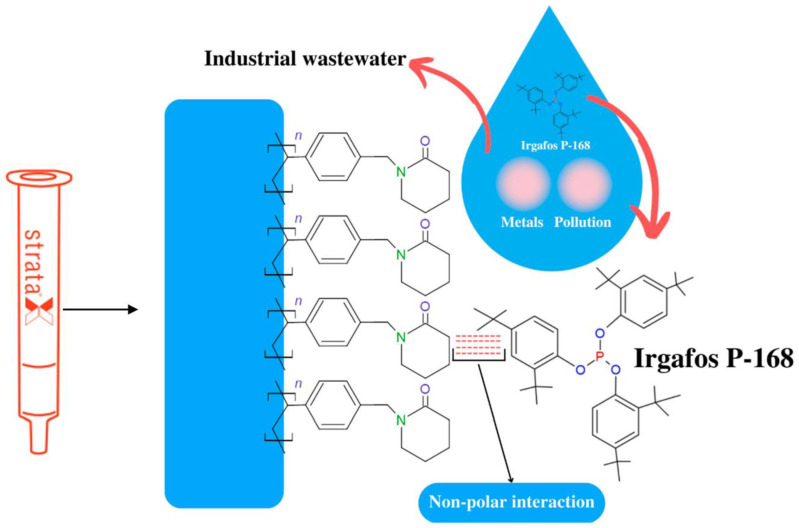
Mechanism of interaction and separation of the Irgafos P-168 in the stationary phase of the SPE.

**Figure 4 molecules-28-03163-f004:**
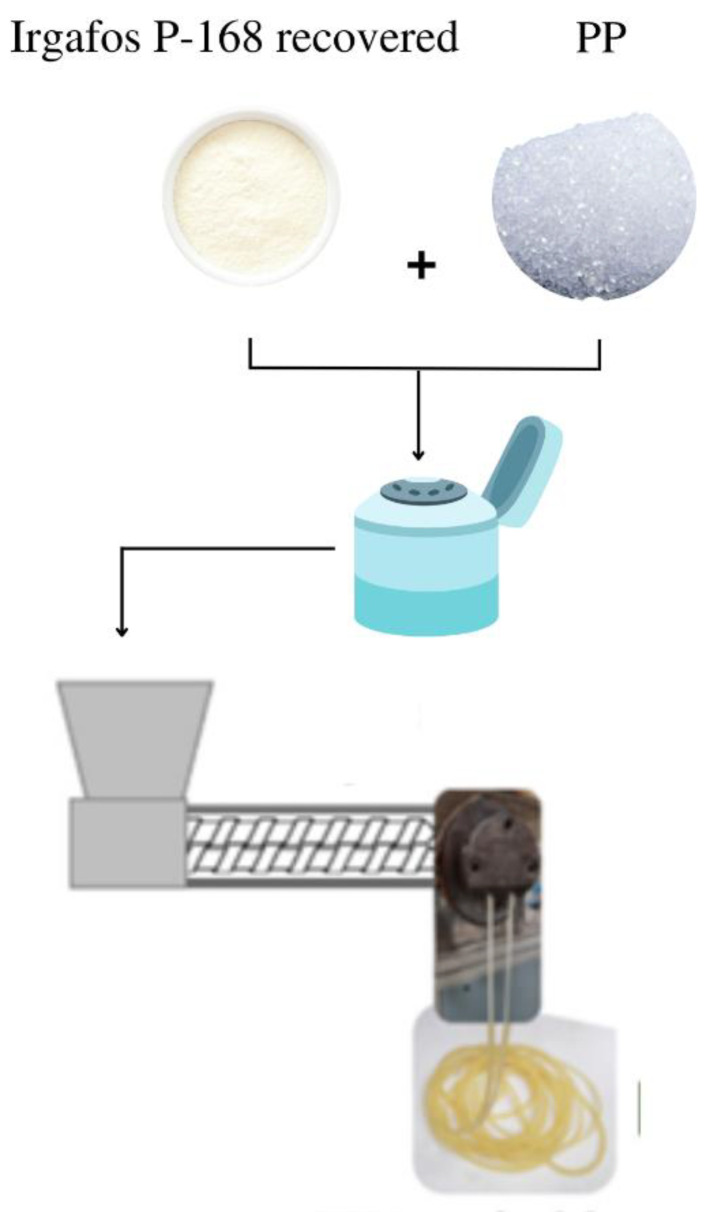
Mixing and extrusion of recovered Irgafos P-168 and pure Irgafos P-168 with virgin PP.

**Figure 5 molecules-28-03163-f005:**
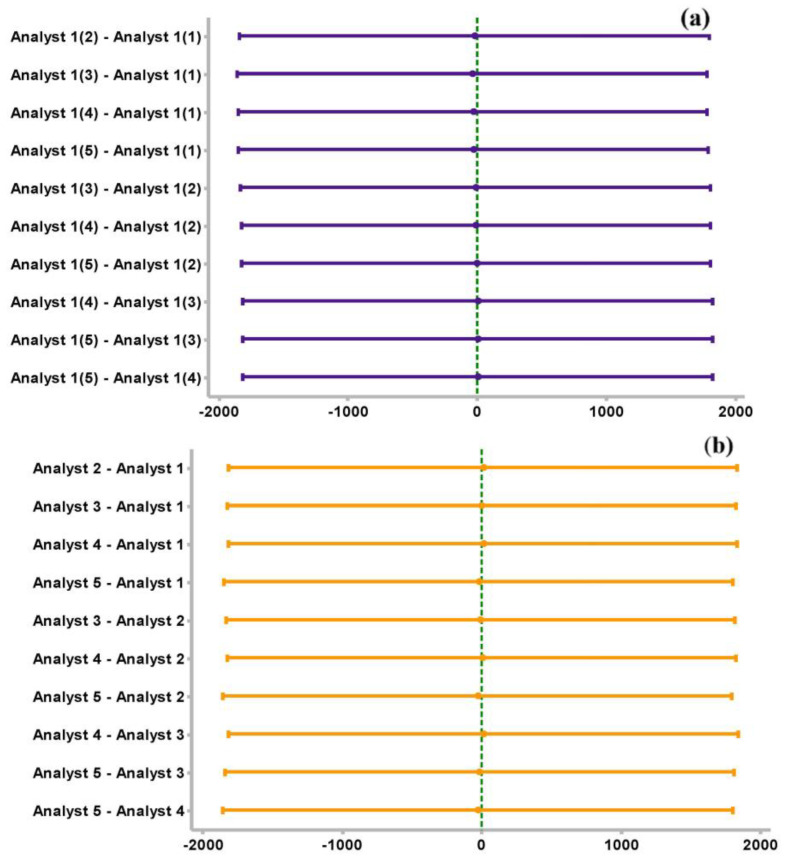
(**a**) Difference of means compared one to one for repeatability of the calibration curve. (**b**) Difference of means compared one to one for calibration reproducibility.

**Figure 6 molecules-28-03163-f006:**
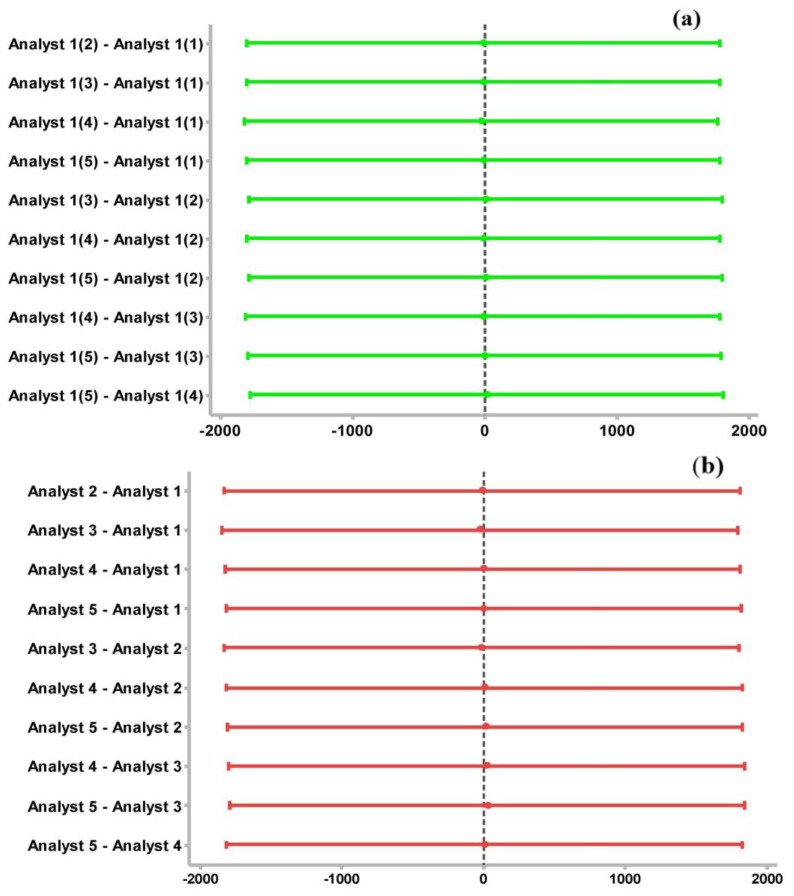
(**a**) Difference of means compared one to one for SPE repeatability. (**b**) Difference of means compared one to one for reproducibility of SPE.

**Figure 7 molecules-28-03163-f007:**
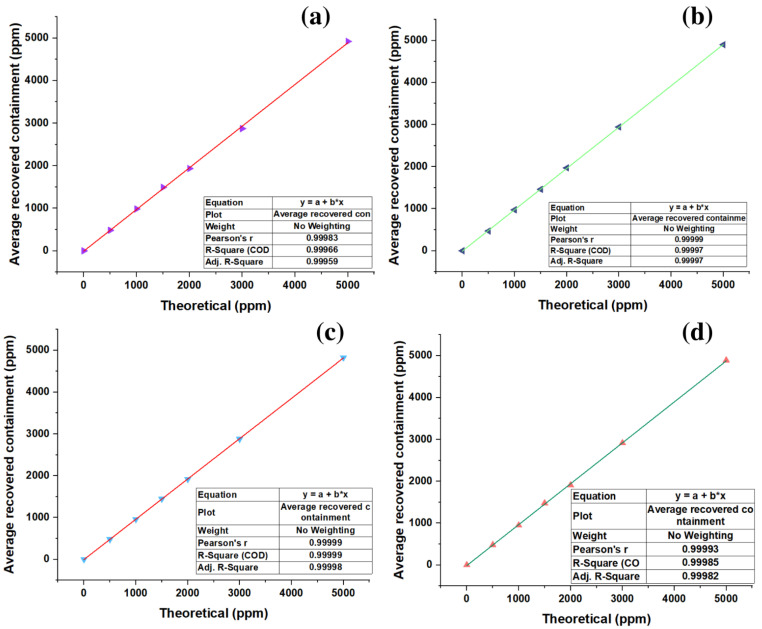
Linearity: (**a**) repeatability in calibration curve; (**b**) reproducibility in calibration curve; (**c**) repeatability for SPE; (**d**) reproducibility for SPE.

**Figure 8 molecules-28-03163-f008:**
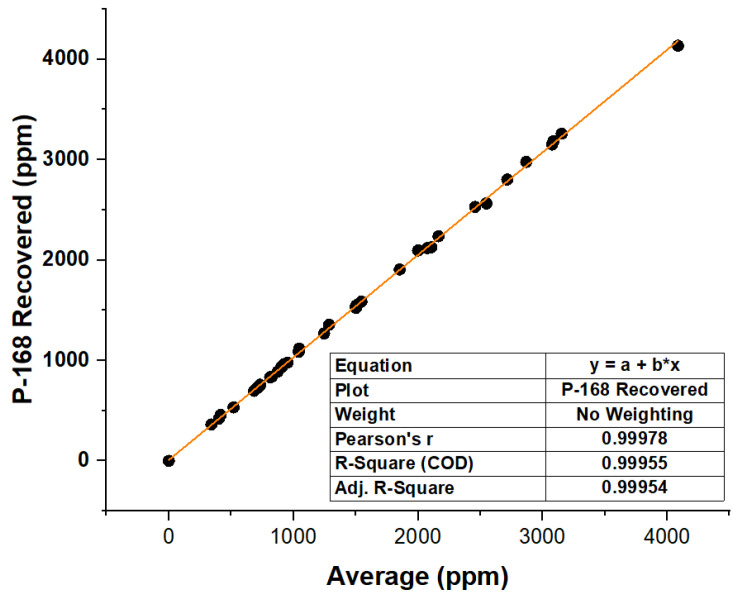
Average SPE concentrations of standards versus average concentration recovered from the samples.

**Figure 9 molecules-28-03163-f009:**
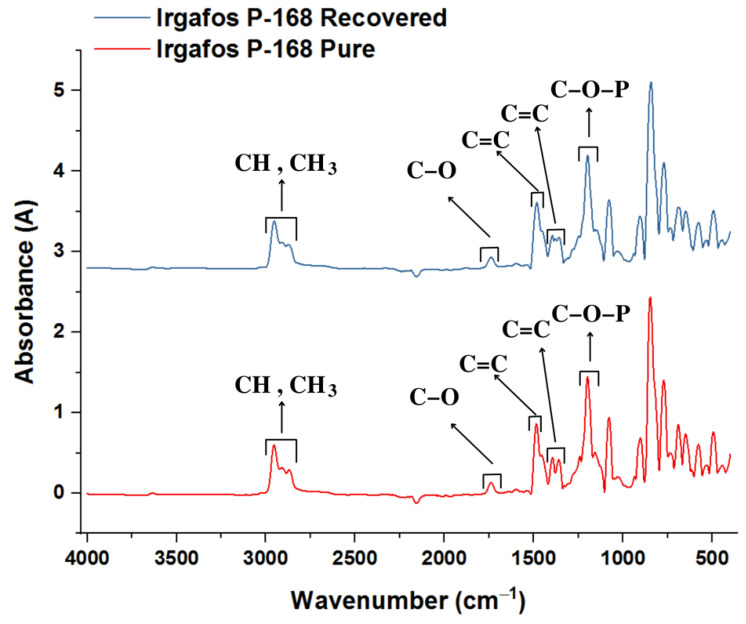
IR spectrogram for pure Irgafos P-168 and IR spectrogram for recovered Irgafos P-168.

**Figure 10 molecules-28-03163-f010:**
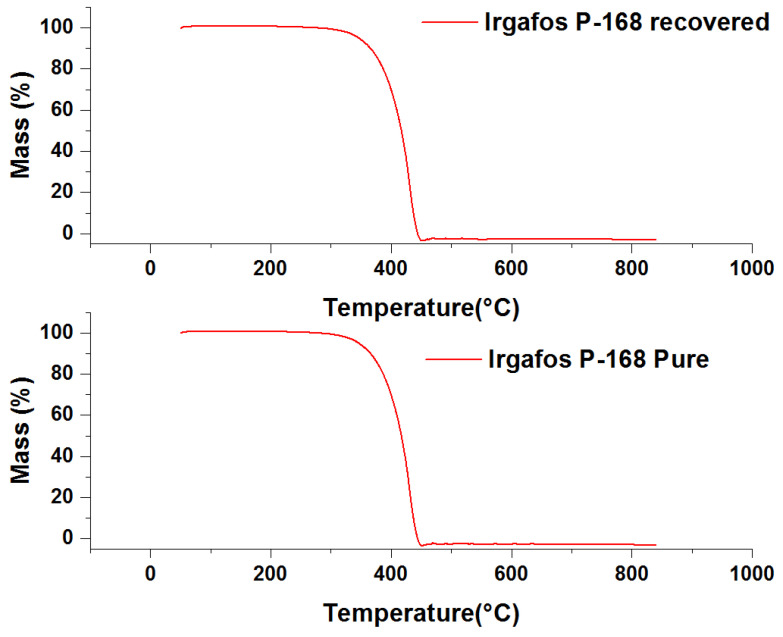
Pure Irgafos P-168 TGA and recovered Irgafos P-168 TGA.

**Figure 11 molecules-28-03163-f011:**
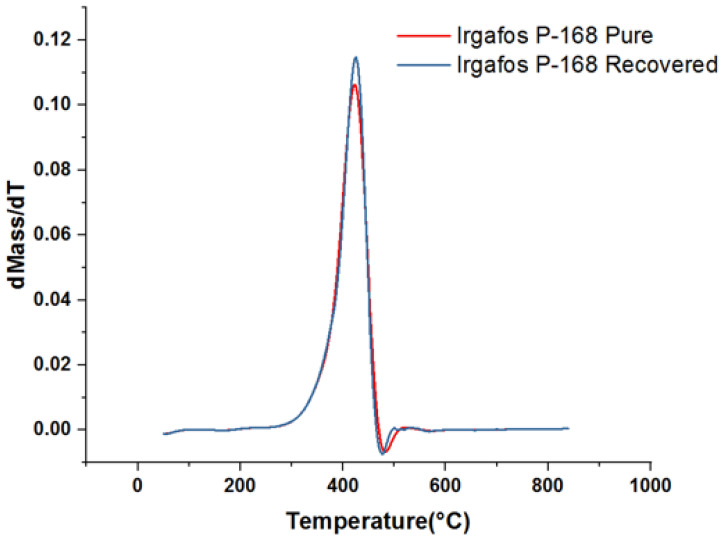
Derivative TGA of pure Irgafos and derivative TGA of recovered Irgafos.

**Figure 12 molecules-28-03163-f012:**
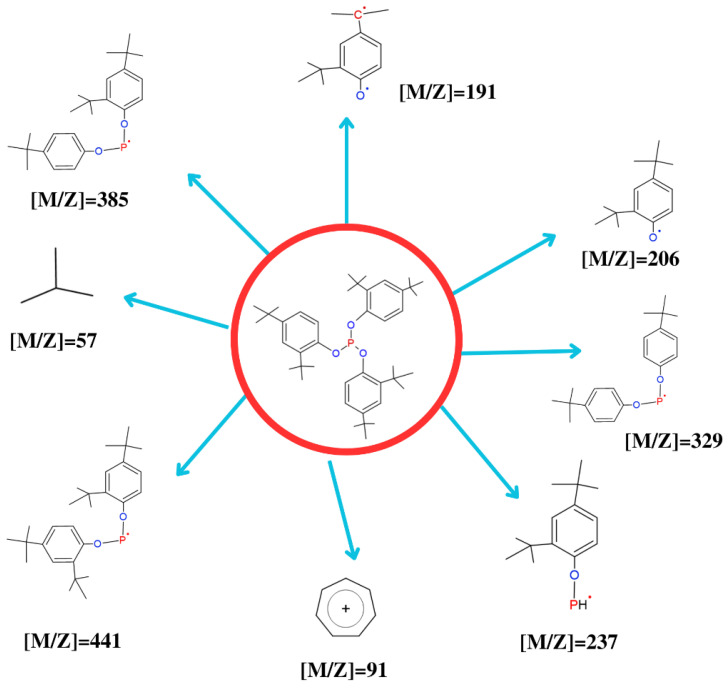
Structure of molecular ions from the fragmentation of Irgafos P-168.

**Figure 13 molecules-28-03163-f013:**
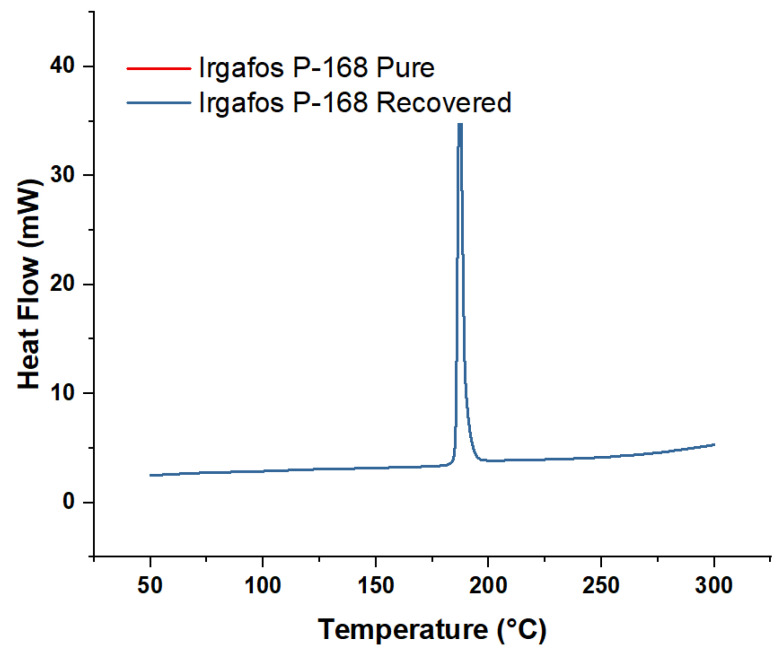
DSC Irgafos 168 pure and DSC Irgafos 168 recovered.

**Figure 14 molecules-28-03163-f014:**
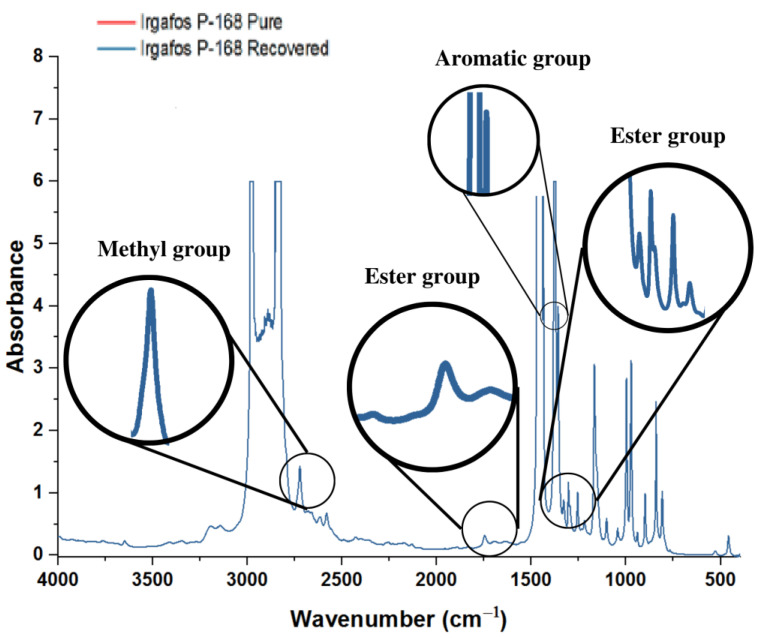
PP + P-168 pure and PP + P-168 recovered.

**Figure 15 molecules-28-03163-f015:**
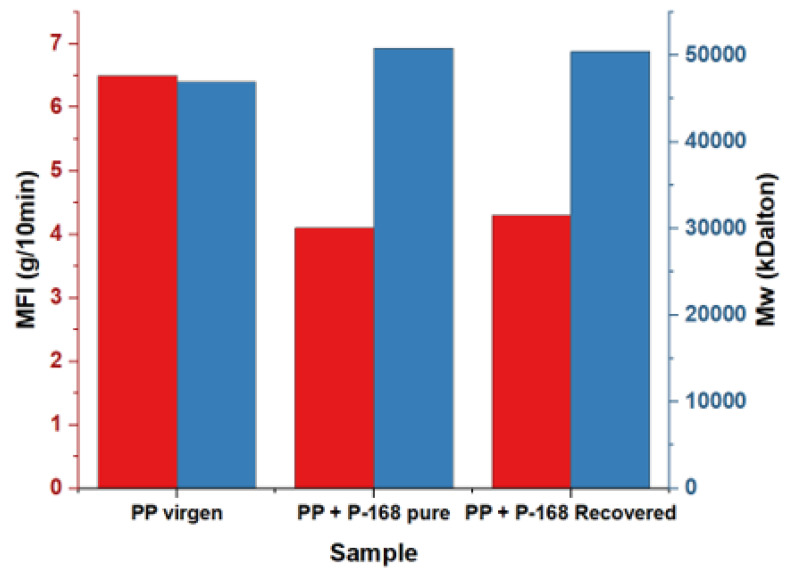
MFI PP + P-168 pure and PP + P-168 recovered.

**Figure 16 molecules-28-03163-f016:**
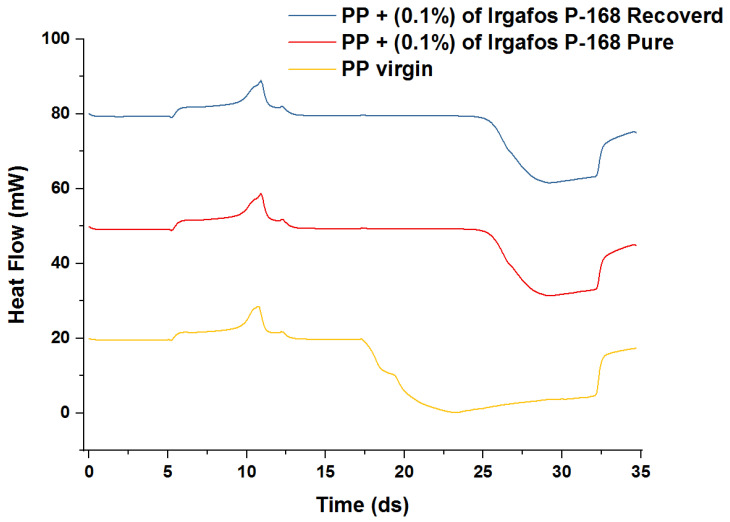
OIT PP, PP + P-168 pure and PP + P-168 recovered.

**Table 1 molecules-28-03163-t001:** Irgafos P-168 repeatability and reproducibility calibration curve data compilation.

Values for Irgafos P-168 Quantitation Model by HPLC-MS							
Sequence Intraday (Same Day)							
Standard	1	2	3	4	5	6	7
**Theoretical**	0	500	1000	1500	2000	3000	5000
**Analyst 1**	0	485	990	1521	1991	2945	4945
**Analyst 1**	0	477	1010	1488	1868	2929	4932
**Analyst 1**	0	488	977	1493	1900	2845	4900
**Analyst 1**	0	490	965	1495	1944	2737	4993
**Analyst 1**	0	475	981	1466	1968	2910	4831
**Average**	0	483.0	984.6	1492.6	1934.2	2873.2	4920.2
**Deviation**	0	6.7	16.8	19.6	50.0	85.1	60.0
**RSD**	0	1.4	1.7	1.3	2.6	3.0	1.2
**Error**	0	3.4	1.5	0.5	3.3	4.2	1.6
**Sequence Interday (different day)**							
**Standard**	**1**	**2**	**3**	**4**	**5**	**6**	**7**
**Theoretical**	0	500	1000	1500	2000	3000	5000
**Analyst 1**	0	443	934	1435	1976	2990	4956
**Analyst 2**	0	456	967	1521	1982	2985	4880
**Analyst 3**	0	482	973	1491	1895	2856	5014
**Analyst 4**	0	495	1011	1399	1989	3011	4876
**Analyst 5**	0	475	986	1458	2013	2867	4779
**Average**	0	470.2	974.2	1460.8	1971	2941.8	4901
**Deviation**	0	20.7	28.1	47.5	44.7	74.1	89.1
**RSD**	0	4.4	2.9	3.3	2.3	2.5	1.8
**Error**	0	6	2.6	2.6	1.5	1.9	2

**Table 2 molecules-28-03163-t002:** Irgafos P-168 repeatability and reproducibility extraction data compilation.

Values for Irgafos P-168 Extraction Method by SPE							
Sequence Intraday (Same Day)							
STDA	1	2	3	4	5	6	7
**Theoretical**	0	500	1000	1500	2000	3000	5000
**Analyst 1**	0	465	933	1396	1943	2950	4911
**Analyst 1**	0	476	969	1477	1922	2854	4795
**Analyst 1**	0	471	972	1482	1896	2889	4811
**Analyst 1**	0	485	985	1396	1958	2756	4824
**Analyst 1**	0	483	932	1484	1876	2949	4779
**Average**	0	476	958.2	1447	1919	2879.6	4824
**Deviation**	0	8.3	24.2	46.6	33.5	80.3	51.5
**RSD**	0	1.7	2.5	3.2	1.7	2.8	1.1
**Error**	0	4.8	4.2	3.5	4.1	4	3.5
**% Recovery**	0	95	96	96	96	96	96
**Sequence Intraday (different day)**							
**STDA**	**1**	**2**	**3**	**4**	**5**	**6**	**7**
**Theoretical**	0	500	1000	1500	2000	3000	5000
**Analyst 1**	0	481	927	1533	1911	2944	4890
**Analyst 2**	0	477	933	1478	1934	2920	4869
**Analyst 3**	0	469	957	1461	1873	2894	4835
**Analyst 4**	0	481	981	1432	1894	2904	4943
**Analyst 5**	0	493	935	1491	1933	2911	4905
**Average**	0	480.2	946.6	1479	1909	2914.6	4888.4
**Deviation**	0	8.7	22.3	37.4	26.1	19	40.3
**RSD**	0	1.8	2.4	2.5	1.4	0.7	0.8
**Error**	0	4	5.3	1.4	4.6	2.8	2.2
**% Recovery**	0	96	95	99	95	97	98

**Table 3 molecules-28-03163-t003:** Quantification of Irgafos 168 in wastewater samples.

Values for the Extraction Method on Samples of Irgafos P-168 by SPE											
Day	Sample	Analyst 1	Analyst 2	Analyst 3	Analyst 4	Analyst 5	Average	Deviation	RSD	Recovery	% Recovery
1	1	1500	1512	1524	1538	1542	1523.2	17.6	1.2	1502	98.61
2	2	2100	2104	2075	2096	2104	2095.8	12.1	0.6	2000	95.43
3	3	1978	1875	1865	1910	1902	1906	44.3	2.3	1853	97.22
4	4	975	963	942	912	925	943.4	26	2.8	901	95.51
5	5	745	702	649	688	700	696.8	34.4	4.9	685	98.31
6	6	462	463	475	435	450	457	15.1	3.3	416	91.03
7	7	412	405	418	435	433	420.6	13.1	3.1	402	95.58
8	8	514	524	546	575	502	532.2	28.9	5.4	520	97.71
9	9	912	975	902	934	942	933	28.5	3.1	912	97.75
10	10	777	800	715	764	738	758.8	33.2	4.4	733	96.6
11	11	1000	975	956	1012	943	977.2	29	3	925	94.66
12	12	1426	1354	1378	1300	1322	1356	49.2	3.6	1286	94.84
13	13	1542	1523	1567	1500	1586	1543.6	34.2	2.2	1508	97.69
14	14	947	977	968	942	975	961.8	16.2	1.7	952	98.98
15	15	871	822	809	839	826	833.4	23.6	2.8	826	99.11
16	16	766	743	709	785	768	754.2	29.4	3.9	733	97.19
17	17	700	705	766	745	735	730.2	27.7	3.8	716	98.06
18	18	974	955	961	908	911	941.8	30.3	3.2	904	95.99
19	19	850	824	809	834	870	837.4	23.6	2.8	813	97.09
20	20	352	342	365	385	366	362	16.2	4.5	342	94.48
21	21	711	702	742	733	746	726.8	19.4	2.7	709	97.55
22	22	911	915	908	977	934	929	28.7	3.1	913	98.28
23	23	1201	1245	1286	1300	1311	1268.6	45.3	3.6	1246	98.22
24	24	1542	1575	1563	1646	1602	1585.6	40.1	2.5	1546	97.5
25	25	2142	2092	2105	2134	2158	2126.2	27.1	1.3	2106	99.05
26	26	2535	2564	2571	2509	2641	2564	49.6	1.9	2548	99.38
27	27	3102	3152	3136	3184	3172	3149.2	32.2	1	3086	97.99
28	28	4150	4135	4106	4172	4108	4134.2	28.1	0.7	4087	98.86
29	29	3012	2975	2942	2955	3001	2977	29.6	1	2869	96.37
30	30	2807	2645	2711	2908	2938	2801.8	125.2	4.5	2716	96.94
31	31	2148	2299	2108	2276	2354	2237	104.4	4.7	2165	96.78
32	32	1545	1506	1562	1575	1536	1544.8	26.4	1.7	1502	97.23
33	33	2015	2106	2185	2153	2137	2119.2	64.8	3.1	2076	97.96
34	34	3102	3245	3371	3262	3312	3258.4	100.3	3.1	3154	96.8
35	35	2571	2534	2516	2509	2511	2528.2	25.9	1	2459	97.26
36	36	3014	3105	3275	3209	3315	3183.6	123.7	3.9	3077	96.65
37	37	1091	1142	1162	1085	1108	1117.6	33.3	3	1046	93.59
38	38	872	877	908	913	873	888.6	20.2	2.3	875	98.47
39	39	1075	1011	1134	1126	1088	1086.8	49.1	4.5	1042	95.88
40	40	1542	1500	1612	1573	1546	1554.6	41.4	2.7	1514	97.39

**Table 4 molecules-28-03163-t004:** Fragment mass and approximate relative abundance of pure and recovered Irgafos.

Qualification Ions of the Irgafos Pure (*m*/*z*)	Qualification Ions of the Irgafos Recovered (*m*/*z*)	Approximate Relative Abundance
57	57	49.0%
91	91	4.8%
147	147	29.8%
191	191	11.8%
237	237	4.7%
308	308	7.8%
329	329	8.0%
385	385	12.1%
441	441	100%
646	646	12.28%

## Data Availability

Not applicable.

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
