# Peer review of "Valoration of the Synthetic Antioxidant Tris-(Diterbutyl-Phenol)-Phosphite (Irgafos P-168) from Industrial Wastewater and Application in Polypropylene Matrices to Minimize Its Thermal Degradation"

_molecules, 2023, doi:10.3390/molecules28073163_

Round 1

Reviewer 1 Report

Attached file

Author Response

Dear

I hope you are doing well and I thank you for taking the time to review this investigation.

I attach my answers.

Can authors add the specific applications of this work in abstract?

Thank you for reviewing this research. We have included the information.

R2 values higher than or at least equal to 0.99955 and correlation coefficient values higher than or at least equal to 0.99978 were obtained. Is that author mean R2 show coefficient of determination?

Thank you for reviewing this research. We have rewritten this sentence for clarity. It refers to the correlation coefficient, that is, to the linearity of the calibration curves.

Figure 6. IR spectrogram for pure Irgafos P-168 and IR spectrogram for recovered Irgafos

P-168. The results of FTIR did not represent clearly can author present Irgafos P-168

spectrum separate in vertically shift?

This is not Fig 6. It is Fig 9. According to your suggestions we have placed the new spectra.

In the Figure 7 cannot find Pure Irgafos P-168 TGA curve.

Thank you for reviewing this research. The corresponding Fig is 10. We have now separated the figures for clarity.

Figure 13. Interacción de Irgafos P-168 con material de retención en capsula de SPE. Is that English? please check Check the sentences and lines grammar.

Thank you for reviewing this research. The corresponding Fig is 3. We have made the language correction.

Reviewer 2 Report

The difference in molecular weight of PP stabilized by pure or recovered Irgafos P-168 is a direct evidence to tell if the recovered Irgafos P-168 has the same effect on thermal stabilization of PP with pure Irgafos P-168. Hence ,GPC measurement of PP with pure Irgafos P-168 or recovered Irgafos P-168 should be carried out.

Author Response

Dear

I hope you are doing well and I thank you for taking the time to review this investigation.

I attach my answers.

The difference in molecular weight of PP stabilized by pure or recovered Irgafos P-168 is a direct evidence to tell if the recovered Irgafos P-168 has the same effect on thermal stabilization of PP with pure Irgafos P-168. Hence ,GPC measurement of PP with pure Irgafos P-168 or recovered Irgafos P-168 should be carried out.

Thank you for reviewing this research. According to your suggestion, now To evaluate the effectiveness of this P-168, we have analyzed the induced oxidation time (this measures the thermo-oxidative stability of the material), the fluidity index (it measures the fluidity of the material and is a measure related to the viscosity of the material) of the virgin PP, the recovered PP+P168 and the pure PP+P168. To determine the molecular weight distribution, we have applied the Brenmer model that allows obtaining results of the molecular weight distribution of the material as a function of its fluidity.

Round 2

Reviewer 1 Report

I recommend accepting this manuscript in present form.

Reviewer 2 Report

This manuscript can be published now.